# Phytoestrogen-Based Hormonal Replacement Therapy Could Benefit Women Suffering Late-Onset Asthma

**DOI:** 10.3390/ijms242015335

**Published:** 2023-10-19

**Authors:** Bettina Sommer, Georgina González-Ávila, Edgar Flores-Soto, Luis M. Montaño, Héctor Solís-Chagoyán, Bianca S. Romero-Martínez

**Affiliations:** 1Departamento de Investigación en Hiperreactividad Bronquial, Instituto Nacional de Enfermedades Respiratorias ‘Ismael Cosio Villegas’, Calzada de Tlalpan 4502, Colonia Sección XVI, Mexico City CP 14080, Mexico; 2Laboratorio de Oncología Biomédica, Departamento de Enfermedades Crónico Degenerativas, Instituto Nacional de Enfermedades Respiratorias ‘Ismael Cosio Villegas’, Mexico City CP 14080, Mexico; ggonzalezavila@yahoo.com; 3Departmento de Farmacología, Facultad de Medicina, Universidad Nacional Autónoma de México, Mexico City CP 04510, Mexico; edgarfloressoto@yahoo.com (E.F.-S.); lmmr@unam.mx (L.M.M.); biancasromero_@hotmail.com (B.S.R.-M.); 4Neurociencia Cognitiva Evolutiva, Centro de Investigación en Ciencias Cognitivas, Universidad Autónoma del Estado de Morelos, Cuernavaca CP 62209, Morelos, Mexico; hecsolcha@yahoo.com.mx

**Keywords:** phytoestrogens, late-onset asthma, hormonal replacement therapy

## Abstract

It has been observed that plasmatic concentrations of estrogens, progesterone, or both correlate with symptoms in asthmatic women. Fluctuations in female sex steroid concentrations during menstrual periods are closely related to asthma symptoms, while menopause induces severe physiological changes that might require hormonal replacement therapy (HRT), that could influence asthma symptoms in these women. Late-onset asthma (LOA) has been categorized as a specific asthmatic phenotype that includes menopausal women and novel research regarding therapeutic alternatives that might provide relief to asthmatic women suffering LOA warrants more thorough and comprehensive analysis. Therefore, the present review proposes phytoestrogens as a promising HRT that might provide these females with relief for both their menopause and asthma symptoms. Besides their well-recognized anti-inflammatory and antioxidant capacities, phytoestrogens activate estrogen receptors and promote mild hormone-like responses that benefit postmenopausal women, particularly asthmatics, constituting therefore a very attractive potential therapy largely due to their low toxicity and scarce side effects.

## 1. Introduction

The scientific literature extensively illustrates that asthma is an inflammatory disease [1,2,3] and, seemingly, symptoms in asthmatic women relate to their sex hormonal status (i.e., perimenstrual period, pregnancy, menopause). In addition, it has been established that most lung diseases are worse in women, a fact reflected in severity, exacerbation rate, hospitalizations, and mortality [4]. Plasmatic concentrations of either estrogens, progesterone, or both have been correlated with asthma symptoms. Female sex steroid hormones and their periodical fluctuations have been considered fundamental in this phenomenon. Alarmingly, about 40% of asthmatic women experience premenstrual exacerbations, around 50% of women hospitalized for asthma episodes are premenstrual [5], and unfortunately, they are more likely to experience corticosteroid refractory asthma [6,7]. Perimenstrual asthma has been characterized as a symptom worsening period and is studied as a pathophysiological entity [8,9,10]. On the other hand, the absence of sex hormonal cycles, i.e., childhood and menopause, favors asthmatic women’s respiratory health, since they are less prone to asthma episode frequency and severity than during other reproductive life stages (puberty, or reproductive maturity). Notwithstanding this, menopause is accompanied with severe physiological changes, including immunological senescence that might require hormonal replacement therapy (HRT) [11,12,13], which could influence asthma symptoms in these women. In addition, late-onset asthma (LOA) has been lately referred to as a specific asthmatic phenotype that includes menopausal women, consequently promoting accelerated and novel research in this regard [14,15,16].

Undoubtedly, women’s hormonal status plays a key role in this illness’ development, severity, and phenotype, which deserve further and more profound studies.

Understandably, research regarding therapeutic alternatives that might provide relief to asthmatic women during these life periods (mainly exacerbations due to perimenstrual asthma or LOA), warrants future clinical studies that corroborate their therapeutic value. In addition, published reports regarding the beneficial effects of phytoestrogens on late-asthmatic menopausal women are practically non-existent. By consulting clinicaltrials.gov online, we found there is currently one study evaluating resveratrol/quercetin in the management of asthma, COPD, and long-lasting COVID (ID NCT05601180) [17], but no results have been published yet. Four clinical trials studying the implementation of isoflavones in asthma are reported: two were completed, one is recruiting study subjects, and one is not yet recruiting (ID: NCT00277446, NCT01052116, NCT00741208, NCT05667701, respectively) [18,19,20,21]. None of these studies encompasses the special interest we discuss in our review. Interestingly, they evaluate inflammatory markers (exhaled nitric oxide, eosinophil LTC4 synthesis, eosinophil count, IL-6, serum CRP, and urinary leukotriene E4/creatinine), pulmonary function (FEV1, FVC, peak flow, morning peak flow rate (PEFR)), symptomology, and asthma quality of life and control questionnaires (Asthma Control Test score, Asthma Symptoms Utility Index score, Marks Asthma Quality of LifeQuestionnaire score, Children’s Health Survey for Asthma score, and Juniper Asthma Control Questionnaire). Conceivably, clinical trials evaluating phytoestrogens effects on late-onset asthmatic women are urgently needed. Therefore, the present review describes how women’s estrogenic hormonal status intertwines with asthmatic inflammation during menopause and revises literature proposing phytoestrogens as a promising adjuvant therapy for these females.

## 2. Female Humoral Immunity Plays a Key Role in Atopic Asthma

In accordance with abundant literature, a considerable proportion of the female world population is prone to allergies and Th2-driven immunological responses [22,23,24]. This fact per se could explain why asthmatic adult women have higher asthma prevalence (the percentage of a population affected with asthma at a given time) than men (9.6% versus 6.3%, respectively) [25,26], but female sex steroid hormones also play a key role in the symptomatology of atopic asthma. It has been corroborated that eosinophilia in patients with child-onset atopic asthma is higher in females than in males [27] and that circulating levels of these inflammatory cells characterize atopic asthma and/or asthma severity [28,29]. Although the contribution of 17β-estradiol (E2) to Th2-biased inflammation has been widely studied, the concentration-dependent effects of E2 remain a tricky aspect in the interpretation of the results. Low concentrations of this hormone likely are immune enhancing, while high concentrations might diminish immunity [30].

Notwithstanding this, it is known that female sex hormones influence the development of dendritic and B cells [31,32], participate in T cell responsiveness [33,34], and modify Th1/Th2 balance [35,36], although some basic immunological aspects remain intriguing. For instance, do females suffer worse asthma symptoms because of a peculiar immunological outset? In this sense, the study by Lauzon-Joset et al. [37] illustrates that female brown Norway (BN) rats have a significantly increased number of immune cells in comparison to males. The augmented number of inflammatory cells appreciated in bronchoalveolar lavages (BAL) included macrophages, neutrophils, eosinophils, and lymphocytes, even though this bias was mostly marked for eosinophils. To further study this issue, they implanted E2-releasing pellets into male BN rats. As a result, they observed that serum E2 concentrations were comparable to female levels and significantly higher compared to untreated males. Furthermore, BAL obtained from E2-treated males had a female-like phenotype, mainly showing an increased number of inflammatory cells compared to untreated males.

Interestingly, the main difference was the eosinophils count, which was higher in comparison to the females. In addition, they observed an increase in neutrophils in E2-treated males, suggesting that E2 administration impacts the common generation of eosinophils and neutrophils. Furthermore, to assess whether these male/female differences were a general phenomenon in rats, they measured the numbers of eosinophils in PVG (Piebald Virol Glaxo) rats’ BAL. As expected, no significant difference in airway eosinophil numbers was observed between males and females, which were also considerably lower compared to the BN strain. In summary, these findings hint to a genetic background linking BN females to Th2 driven immunological responses. Conceivably, the obvious question would be if these phenomena could happen in humans. In this regard, some authors [37] claim that similar circumstances were seen in studies contrasting the asthma-resistant Amish vs. asthma-susceptible Hutterites [38], amongst whom the most affected subjects are females [39], and conclude that the Th2-enhancing effects of E2 depend on an elevated baseline Th2 bias as observed in the BN strain and in atopic individuals. Furthermore, the former findings may reflect an actual circumstance, since it has been reported that atopy-associated asthma symptoms have augmented in females but not males since the 1990s [39], indicating that a female-specific factor influences the disease development, closely duplicating the immunological particularities seen in BN rats.

## 3. Late-Onset Asthma

In contrast to the well-studied atopic asthma characteristic of the childhood onset illness, late-onset asthma presents symptoms for the first time during adulthood, and Miranda et al. [40] suggested differentiating asthma into early versus late onset based on the age of symptoms development and presence of airway eosinophilia. Indeed, this phenotype is non-allergic, shows the presence or absence of eosinophilic inflammation, and has a significantly higher incidence (rate of occurrence of new asthma cases in a population) in women [40,41,42] (Figure 1). This incidence appears to be 4.6 cases per 1000 persons in females and 3.6 in males, with a tendency to increase with age [43]. According to the literature [40,44] late-onset asthma begins when patients (mostly women) start showing symptoms at around 27 years of age on average, but LOA might develop any time from 27 years of age onwards, even though much controversy remains in this sense [44,45]. The prevalence of asthma in adults older than 65 years is as high as 10%, and females are the main constituents of the 64–75 age group [44]. In this sense, observations derived from a population of 9091 males and females followed for 8–10 years report that asthma was 20% more frequent in females than in males over the age of 35 years at the beginning of the study, while non-asthmatic subjects showed a higher incidence of asthma in females than in males. Surprisingly, more than 60% of females and 30% of males with new-onset asthma were non-atopic, and this trend was observed from puberty to menopause in women, and no difference in the incidence of allergic asthma between sexes was observed [42]. Moore et al. documented that late-onset, female-predominant asthma phenotype mainly consists of women ranging from 34 to 68 years of age and mostly atopic, with high body mass index and decreased baseline pulmonary function [46]. Other authors identified two female-specific asthma phenotypes among patients older than 20 years of age: those with atopy and eosinophilia and those with obesity and neutrophilic inflammation [47] (Table 1). Although some research indicates that the incidence of asthma lowers after menopause [48], other sources report increased frequency of LOA in women during this life stage [49]. It is noteworthy that asthma prevalence and incidence have been observed to decrease among reproductive-aged women using contraceptives [50,51,52,53], and lately, encouraging data point out that hormonal replacement therapy (HRT) might prevent LOA in post-menopausal women [54].

## 4. Hormone Replacement Therapy Pros and Cons

Indeed, HRT seems to significantly reduce the probability of developing LOA in menopausal women [54]. Regarding these women, it has been reported that HRT increases pulmonary function parameters like forced expiratory volume in one second (FEV1), forced vital capacity (FVC), forced expiratory flow 25–75%, and peak expiratory flow rate, and favors less airway obstruction and hyperresponsiveness [59,60,61,62]. It has also been noticed that the risk of new-onset asthma in postmenopausal women is at its highest in the transitional period from early postmenopausal to the late postmenopausal state, although circumstantial conditions might contribute to conflicting results in the evaluation of menopause relation to LOA and the benefits that HRT might provide [14,54]. Notwithstanding this, probable HRT side effects include cardiovascular events [63], thromboembolic disease [64], stroke [65], and breast cancer [66]. Herein, we suggest that phytoestrogens might be a beneficial HRT alternative with less side effects than conventional hormonal therapies. Because phytoestrogen-based HRT has been scantly investigated, some aspects like their role in LOA remain uncertain and deserve a critical review.

## 5. Menopause Is an Inflammatory State

Sexual dimorphisms in body weight, food intake, glucose/lipid homeostasis, and insulin sensitivity have been related to E2 metabolic functions [67,68,69,70,71], and as expected, the decline of circulating E2 concentrations due to menopause induces significant changes in metabolism, fat distribution, inflammation, and insulin activity [72,73,74,75]. Conceivably, menopause symptoms are related to these metabolic alterations, and although HRT is a viable treatment option to alleviate the symptoms of menopause [76], it is associated with oncogenic and cardiovascular risks [77]. Notwithstanding this, it has been observed that E2 supplementation in mice increases the expression of antioxidant enzymes and reduces inflammation [78,79,80]. Seemingly, decreases in plasmatic E2 concentrations during menopause predispose to inflammation characterized by increased levels of interleukins IL-1β, IL-6, and IL-8, as well as of tumor necrosis factor-α (TNF-α), IL-4, IL-10, and IL-12 [55,56,57,58] (Table 1). In general terms, older asthmatics show systemic inflammation that closely reproduces the severe phenotype observed in younger patients: high neutrophil blood counts, and augmented concentrations of IL-6, IL-8, and C-reactive protein [81,82,83]. Conceivably, age-related changes in hormonal status, innate and adaptive immunology, and systemic inflammation may predispose elderly people to increasing rates of infections and consequent exacerbated asthma, but most interestingly, they might be primordial factors in the initiation of LOA [84].

## 6. Characteristics of Phytoestrogens

Phytoestrogens are compounds found in plants that resemble estrogens in their molecular structure and size, particularly E2. Because of these structural characteristics, they can exert estrogenic and/or antiestrogenic effects [85]. Such effects are concentration-dependent phenomena; when phytoestrogens are present at an adequate concentration, they generate estrogenic consequences, and if the concentrations are high, the result will be antiestrogenic [86].

On the other hand, phytoestrogens are plant polyphenols, a group that includes lignans and isoflavones. Fundamentally, polyphenols are metabolites that confer protection on plants against pathogens and ultraviolet radiation [87] and provide humans with health benefits that depend on their chemical structure (i.e., diverse polyphenol types theoretically award characteristic health benefits [88]). When consumed, lignans are metabolized by anaerobic bacteria in the human gut to enterolignans (also called mammalian lignans) that present steroid-analogous chemical structures and therefore are recognized as phytoestrogens with confirmed estrogenic activity [89]. Isoflavones resemble estrogen in structure and are therefore also classified as phytoestrogens [90]; the latter are further described below.

## 7. Isoflavones and Their Actions in Hormone Replacement Therapy

Some well-studied isoflavones are genistein, daidzein, glycitein, biochanin A, and formononetin; some metabolites of these flavonoids, like equol, which derives from daidzein, also have important physiological activities. The main dietary sources of isoflavones are soybean (*Glycine max*), which contains significant amounts of daidzein, genistein, and glycitein and red clover (*Trifolium pratense*), which comprises formononetin and biochanin A.

In a natural setting, phytoestrogens act as phytoalexins, i.e., low-molecular weight complexes responsible for fungistatic, antibacterial, antiviral, and antioxidant activities in plants [91]. Conceivably, these natural settings, i.e., humidity, temperature, soil type, etc., greatly determine the amounts of phytoestrogens produced by a certain plant; adverse conditions (e.g., poor humidity, pathogens, or plant disease) dramatically increase isoflavone synthesis. Finally, isoflavone plant concentrations are subjected to post-harvest practices like storage and drying [92,93]. In the human diet, isoflavone main sources are soy and soy-derived products. It is known that soybeans contain about 1.5 mg/g isoflavones, and this content is lower in soy-derived foods [94]. Meanwhile, red clover is a component of human dietary supplement food and pharmaceutical products used to reduce menopausal symptoms in women [95]. In this regard, red clover phytoestrogens have become an effective alternative HRT [96]. Furthermore, results obtained from isoflavone studies performed on various subjects suggest that microflora involved in the metabolism of isoflavones influences their ultimate effect on the organism because it transforms the initial molecules into metabolites, such as equol, with modified (augmented) estrogenic activity. In vitro studies proved that equol is more estrogenic [97], and a better antioxidant [98] than daidzein. It is worth mentioning that it is produced by specific colonic microflora (i.e., *Streptococcus intermedius* ssp., *Ruminococcus productus* spp., and *Bacteroides ovatus* spp. [99]) and that, as expected, the ability to generate it relies on such intestinal flora composition. Inoculation of germ-free rats with human flora from equol producers provides the animals with the capability to produce it [100], notwithstanding that an immense interindividual variability among humans in equol production has been reported. Only 30–50% of the occidental people, called “equol producers”, generate significant equol quantities after isoflavone consumption [101,102]. Interestingly, some sources establish that equol estrogenic properties are comparable to those of the original isoflavone [101,103] while others claim that it is pharmacologically important, since it is more estrogenic than daidzein [104].

Meanwhile, genistein’s capacity to reduce menopausal symptoms remains unclear [105,106]. Nevertheless, it has been shown that dietary supplements that contain isoflavones reduce hot flashes frequency in around 10–20%. Understandably, stronger isoflavone activity was more evident in women with more frequent flashes [107]. In women in the reproductive age, isoflavones may cause menstrual cycle disorders (dysmenorrhea), endometriosis, and secondary infertility [108], symptoms that mostly disappear with a soyabean-free diet.

## 8. Resveratrol Is an Agonist for the Estrogen Receptor

Although not a soya bean product, resveratrol has lately attracted attention as another source of potential therapeutics. It is known that it binds to the estrogen receptor and is therefore considered a phytoestrogen [109]. Resveratrol is a polyphenol that acts as a phytoalexin present in grapes, mainly in their skin, and therefore, wine contains significant amounts of this substance [110,111,112], a fact that has been related to the benefic cardiovascular consequences of this beverage’s consumption [113,114,115]. On the other hand, it has been reported that it exerts its anti-inflammatory effects via estrogen receptor-independent pathways [116,117]. Long-term resveratrol treatment prevents ovariectomy-induced osteopenia.

Regarding resveratrol’s potential as HRT, a study carried out in rats compared the effects of these polyphenol vs. estradiol valerate (EV) administration. Ovariectomized (OVX), resveratrol-treated rats’ femoral bone mineral density was significantly higher than from the OVX group and comparable to OVX EV-treated rats. In the OVX group, resveratrol significantly attenuated the increase in urinary Ca, P, as well as seric IL-6 and augmented alkaline phosphatase concentrations. Uterine atrophy observed in the OVX group was overcome with EV treatment, whereas resveratrol did not show any effect [118].

Meanwhile, the Resveratrol for Healthy Aging in Women (RESHAW) trial, a 24-month randomized, double-blind, placebo-controlled study, showed in postmenopausal women that the administration of 75 mg resveratrol twice daily augmented bone density in the lumbar spine and neck of the femur. Resveratrol benefits on bone density were greater in women supplemented with vitamin D plus calcium [119]. These encouraging data warrant further research in resveratrol’s effectiveness as HRT in climacteric women. For more detailed information about this phytoestrogen see Qasem RJ [120].

## 9. Phytoestrogens Mode of Action

Although not their only way of action, it is known that phytoestrogens exert many effects through estrogen receptor (ER) occupancy. Because of the frequent presence of genetic polymorphisms in the ER [121] upon which they act, they may also be considered endocrine disruptors with possible negative influences on the state of health in a certain part of population [122]. Indeed, although the chemical structure of isoflavones is different from that of endogenous estrogens, they can attach and activate the ER [123]. This receptor presents two intracellular isoforms, α-ER and β-ER [122], that act like nuclear factors provoking genetic effects. Isoflavone affinity for the β-ER is approximately five times higher than its affinity for the α-ER [124]. Meanwhile, E2 affinity for both receptor types is basically identical [125]. These types of ERs are unevenly distributed in diverse tissues: β-ER is located predominantly in the bones, lungs, prostate, urinary bladder, skin, and brain, while α-ER is situated in the mammary gland, testes, uterus, kidneys, lungs, and hypophysis [126]. Notably, the literature sustains that ERs expression differs between asthmatics and non-asthmatics. Seemingly, these receptors, primordially β-ER, are upregulated in asthma or during inflammation [127]. Furthermore, it has been demonstrated that pharmacological activation of the β-ER diminishes human airway smooth muscle proliferation in vitro [128] and in a mice asthma model in vivo, where it also lowered airway hyperresponsiveness [129]. Meanwhile, β-ER KO mice showed weakened lung function at baseline in comparison to wild-type (WT) and α-ER KO mice, and the worst changes were observed in females, a fact contributing to findings regarding them as more prone to develop asthma. Contrastingly, α-ER KO mice of either sex had a normal lung function comparable to WT mice at baseline [128,129]. When these mice (WT, α-ER KO, and β-ER KO) were sensitized and challenged, they showed a significant decline in lung function, with the most prominent detriment seen in female mice compared to males of the same group [130]. This finding could be fundamental to explain the abundant clinical data documenting increased severity of asthma in females.

In addition to the α and β ERs-induced genomic responses, E2 can also prompt rapid non-genomic signaling through the G-protein-coupled estrogen receptor (GPER, previously known as GPR30 [131,132]), initially considered an orphan receptor [133]. It has been found that GPER’s long-term activity might lead to gene transcription as well [134,135,136]. GPER signals through many G proteins, including Gαs [137,138] and Gαi [139,140] proteins, and through Gβγ [141] and Gαq/11 [142]. On the other hand, its signaling has been related to epidermal growth factor receptor transactivation [141]. These receptors can be activated by phytoestrogens such as genistein [143], daidzein [144], equol [145], resveratrol [146], and others, but further study is needed to define their possible role in health or disease. Interestingly, among the GPER ligands, only G-1, a nonsteroidal, high-affinity, highly selective agonist [147] has so far entered clinical trials, showing auspicious results as antitumor agent [148,149,150]. On the other hand, it has been found that GPER participates in eosinophilic apoptosis [151], a characteristic that might become an attractive therapeutic target in atopic asthma and other allergic ailments. In this sense, it has been confirmed that G-1 possesses encouraging therapeutic potential for asthma treatment, since it diminished airway hyperresponsiveness and inflammation (decrease in IL-5 and IL-13 levels in bronchoalveolar lavage fluid) in asthma models. Furthermore, it was also corroborated that its use increases regulatory T cells and their production of the anti-inflammatory cytokine IL-10 [152]. The generation of IL-10 by this treatment has also been reported in Th17 cells [153,154]. In surgically postmenopausal mice, chronic treatment with G-1 reduced TNF, MCP1, and IL-6 concentrations and lowered the expression of inflammatory genes [155]. These effects were more noticeable in females, and were absent in Gper-deficient mice, thus demonstrating the selectivity of G-1 [156].

## 10. Phytoestrogens as HRT in Women with LOA

Phytochemicals constitute an attractive source for novel asthma therapies mostly due to their low toxicity and scarce side effects. Their outstanding anti-inflammatory capacities include antioxidant properties [157] and regulation of the inflammatory/immunological cellular setting [158,159,160]. In the context of asthmatic patients, regarding the role played by 6-shogaol (a volatile phenolic compound and ginger constituent which imparts a pungent and spicy-sweet fragrance) [161] in enhancing β_2_ adrenergic receptor agonists’ effect, some authors [162,163] point out that it might become a therapeutic tool with remarkable potential. Besides possessing some of these properties, phytoestrogens also activate estrogen receptors, unleashing mild hormone-like responses that benefit postmenopausal women, particularly asthmatics. In this regard, one research investigated in an in vitro model if genistein and resveratrol had any effect on the cytokine production pattern of splenocytes, and the results demonstrated an increased IL-10 production [164]. It has been documented that IL-10 induces tolerance to allergens [165,166,167] and favors allergic inflammation termination [168,169]. Incidentally, decreased IL-10 production has been observed in patients with severe asthma [170,171]. In addition, there are data revealing a reduction in the inflammatory molecules IL-6 and TNF-α in women’s plasma levels after they received nutritional supplementation with isoflavones [172,173,174]. Chi et al., using soy isoflavones at a dose of 90 mg/day for 6 months, noted a reduction in the levels of IL-6 and TNF-α [172]. Moreover, Nadadur et al., also observed a reduction in levels of TNF-α, but no effect on IL-6 levels using food supplementation with either 50 mg isoflavones or 15 g soy protein in the form of tofu for 8 weeks [173]. Another study with 80 mg isoflavones (60.8 mg of genistein, 16 mg of daidzein, and 3.2 mg of glicitein) for 6 months showed reductions in TNF-α levels [174].

## 11. Phytoestrogens Potential in Pulmonary Fibrosis Treatment

The therapeutic potential of phytoestrogens has been investigated in other pulmonary ailments as idiopathic pulmonary fibrosis (IPF), an inflammatory disease characterized by fibrotic phenomena (epithelial to mesenchymal transition, extracellular matrix production, and collagen formation) that develop in the lungs. Relevantly, IPF incidence has increased in recent years [175], urging the development of novel therapeutic tools. In this sense, Andugulapati et al. [176] investigated the anti-fibrotic properties of Biochanin-A (BCA; a phytoestrogen isolated from the red clover *Trifolium pratense* L. used to relieve postmenopausal discomfort in women) against TGF-β-mediated lung fibrosis in a rat model of IPF. Although further research is needed, BCA treatment showed promising potential for the future IPF treatment, since it significantly diminished the expression of TGF-β fibrotic genes-modulated protein expression and notably reduced inflammatory cell-infiltration, expression of inflammatory markers, and collagen deposition in lung tissues. On the other hand, it has been postulated that consumption of higher amounts of dietary phytoestrogens might be associated with lower IPF prevalence. To confirm this assumption, Solopov et al. [177] studied the effects of dietary phytoestrogen (content of isoflavones 150–340 mg/kg of rodent commercial pellets), on a mice IPF model generated by 0.1 N HCl. They measured IPF severity through lung function, bronchoalveolar lavage fluid, and lung tissue, and found that mice on a phytoestrogen-free diet had increased mortality and worse IPF signs observed as a higher expression of collagen, extracellular matrix deposition, histology, and lung mechanics, and suggested that phytoestrogens may be helpful constituents of a therapeutic program against lung fibrosis. In another study, Zhao et al. [178] tested formononetin-7-sal ester (FS; a synthetic derivative of the phytoestrogen formononetin found in red clover) anti-pulmonary fibrosis potential. Their results showed that FS effectively prevented proliferation and migration of mouse lung fibroblast (cell line L929) stimulated with TGF-β1, and reduced lung fibrosis in a bleomycin-induced pulmonary fibrosis model in mice. Even though much research is still required, all these findings point out that phytoestrogens might also have vast therapeutic potential in IPF treatment.

## 12. Final Considerations

Any disease is an extremely complex entity that can seldom be recognized by a single symptom. LOA seemingly develops in adulthood, a feature shared by eosinophilic granulomatosis with polyangiitis (EGPA, also known as Churg–Strauss syndrome), a rare disease characterized by adult-onset asthma, blood and tissue eosinophilia, and small-vessel vasculitis [179]. EGPA develops mostly in adult patients (mean age of onset around 50 years [180]) but can also affect children [181] and shows no gender dominance [182]. Asthma is mostly the first disease manifestation and appears in 95–100% EGPA patients [183,184]. EGPA shows eosinophilia in sputum and negative allergy tests [185]. Plausibly, these symptoms can also be found in some patients with severe adult-onset asthma, and therefore, LOA has been considered a stage before EGPA development by some authors [186]. Up to now, no definite initiating agent or mechanism has been described for EGPA, but glucocorticoids are the principal treatment option, and some biological therapies have proven to be effective as for instance rituximab (a chimeric monoclonal antibody against the CD20 antigen causing B-cell depletion [187]), omalizumab (an anti-IgE monoclonal antibody that prevents IgE-mediated degranulation of eosinophils [188]), mepolizumab (a monoclonal antibody that prevents the binding of IL-5 to its receptor, hindering eosinophil maturation and survival [189]), and some others like benralizumab and reslizumabare (anti-IL-5α receptor monoclonal antibodies targeting IL-5 axis) [190] that are still under investigation. Notwithstanding this, and according to most available sources, EGPA therapy closely resembles asthma treatment and, consequently, should follow the pharmacological approach recommended by the International Global Initiative for Asthma (GINA) [191]. Further and detailed information regarding EGPA can be found in Fijolek and Radzikowska [192]. Finally, and in contrast to LOA, EGPA has shown no gender dominance [182]; therefore, it seems unlikely that phytoestrogens might have any effect on this illness.

## 13. Conclusions

Although much more research is needed, these novel therapies have a surplus advantage as they might also overcome some menopause symptoms with fewer side effects than existing treatments. They may also contribute to more efficient responses to infection and inflammation leading menopausal women to a much better quality of life.

## Figures and Tables

**Figure 1 ijms-24-15335-f001:**
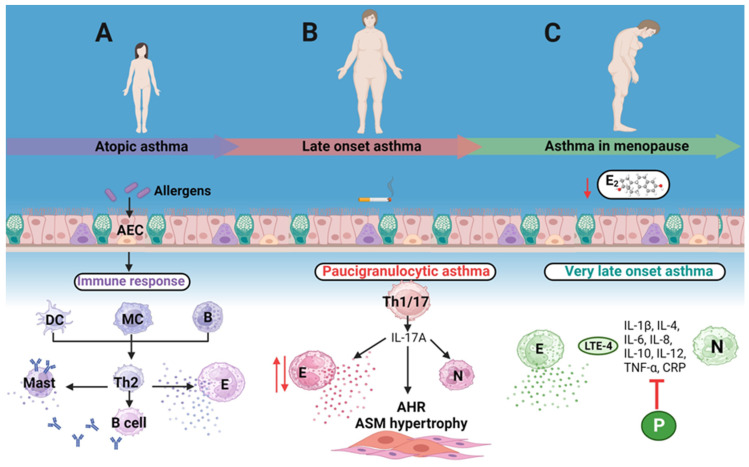
Different asthma phenotypes in females. (**A**) Atopic asthma characteristic of childhood onset illness. (**B**) Late-onset asthma favors a Th1/17 response over a Th2 response, is associated with obesity and pollutants such as cigarette smoke, is non-allergic, and mostly shows absence of eosinophilic inflammation (paucigranulocytic asthma, red arrow pointing down) although sometimes increased eosinophilia is seen (red arrow pointing up) and presents airway smooth muscle remodeling. It has a significantly higher incidence in women and develops between 27 ± 1.3 years (mean age at onset) and 65 years of age. (**C**) Very late onset asthma develops in females of ≥65 years of age and is closely related to the lack or very low concentrations of circulating estrogens. Inflammation related to the absence of estrogenic hormones might be diminished by phytoestrogens. AEC: airway epithelial cells. DC: dendritic cell. MC: macrophage. B: B cell. Mast: mast cell. E: eosinophil. N: neutrophil AHR: airway hyperresponsiveness. ASM: airway smooth muscle. E_2_: 17β-estradiol. LTE-4: Leukotriene E4. CRP: C reactive protein. P: Phytoestrogen.

**Table 1 ijms-24-15335-t001:** Comparison between atopic and late-onset asthma inflammatory characteristics.

Atopic Asthma	Late-Onset Asthma (LOA)	Factors Contributing to LOA during Menopause
- Eosinophilia - Th2 immune response - Th17 associated with severity - Modified Th1/Th2 balance - Higher levels of macrophages, eosinophils, and lymphocytes	- Non-allergic phenotype, with or without eosinophilia - Obesity and neutrophilia	- Changes in metabolism, fat distribution, inflammation, and insulin resistance - E2 low levels predispose to increased levels of IL-1β, IL-4, IL-6, and IL-8, IL-10, IL-12, and TNF-α [55,56,57,58]

## Data Availability

Not applicable.

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
