# Peer review of "Phytoestrogen-Based Hormonal Replacement Therapy Could Benefit Women Suffering Late-Onset Asthma"

_ijms, 2023, doi:10.3390/ijms242015335_

Round 1

Reviewer 1 Report

The manuscript of Sommer and co-authors is well-written and dedicated to the effectiveness of natural flavonoids as a hormone-replacement therapy against bronchial asthma. Indeed, phytoestrogens showed promising effects against a number of chronic lung diseases, especially during natural estrogen deficiency.

As a minor comment, I would suggest adding a short discussion of recent studies about the effectiveness of phytoestrogens in other chronic pulmonary diseases such as pulmonary fibrosis and lung granulomatosis. 

Author Response

October 6th, 2023

REVIEWER 1

As a minor comment, I would suggest adding a short discussion of recent studies about the effectiveness of phytoestrogens in other chronic pulmonary diseases such as pulmonary fibrosis and lung granulomatosis. 

Authors are grateful for the reviewer´s suggestion and added two paragraphes at the end of the amended manuscrit (pages 8-9, lines 354-407). We hope we complied with the reviewer´s expectancies.

Reviewer 2 Report

The review “Phytoestrogen-based hormonal replacement therapy could benefit women suffering late onset asthma” aims to investigate if phytoestrogen is an effective treatment against late onset asthma in women. The aim of the study is recommendable and interesting. However, the aim is not addressed properly in the text.

Major concerns:

1: The manuscript is very unfocused. The aim of the review was to investigate whether phytoestrogen-based therapy decreases asthmatic symptoms in late onset asthmatic women. The main focus of the text should then be to investigate the prevalence of this patient group, what characterizes it and separates it from other groups, and if phytoestrogens would be beneficial by presenting clinical trials with these end points. Investigating where phytosterols are found (foods, beverages), whether the active compounds are formed by the intestinal flora, what receptors are activated, confuse the point. Please revise the text to be more focused on the aim and less on other things, which of course are interesting, but do not further the argument.

Line 339: They may as well contribute to more efficient responses to infection and inflammation leading menopausal women to a much better quality of life.

This sentence illustrates the issue above. The aim was to investigate if phytoestrogen would be beneficial against asthma, not if it effective against infections or general inflammation or give better quality of life to menopausal women.

If you want to argue that phytoestrogens are a good treatment for asthma, you must show clear clinical evidence that this is the case. Explaining estrogen effects on transcription and GPCRs will not further the point.

The manuscript must be focused on the aim, or the aim must be changed.

2: Incorrect statements, for example:

Line 67: circulating levels of these inflammatory cells characterize atopic asthma and/or asthma severity

Is that strictly true? Are levels of circulating eosinophils directly correlated with asthma severity?

Line 120: this trend was observed throughout all the reproductive years

Please explain how “reproductive years” can be compared between males and females? Women reach menopause, men do not.

Minor concerns:

1: There are many language errors in the text, for example:

First sentence of the abstract and line 37: what is “plasmatic concentrations”? Do you mean plasma concentration?

Female sex steroid hormones periodical fluctuation

Not English, must be rephrased.

relief for both their menopause

What is meant by that expression? How can you receive relief for a menopause? You can receive relief from symptoms due to menopause, but not from the menopause itself.

Line 35: symptoms in asthmatic women relate to their sex hormonal status

Line 45: asthma episodes frequency: please remove the final s in episodes.

2: Expressions with questionable meaning, for example:

warrants more thorough and comprehensive analysis.

The expression is too vague. Please specify what should be analyzed.

This statement is wrong. Women do not suffer from asthma just because they are women. Please rephrase.

Line 61: a considerable proportion of the female world population is prone to allergies and Th2-driven immunological responses

This is not limited to the female population.

Line 63: explain why asthmatic adult women have higher asthma prevalence than men (9.6% versus 6.3%, respectively)

What do you mean by prevalence? Do they have more exacerbations?

Line 112: It has been proposed that LOA develops between 27 ± 1.3 years (mean age at onset) and ≥65 years of age

What do you mean by this statement? Please clarify.

Line 114: The prevalence of asthma in adults older than 65 years is as high as 10%, and females are the main constituents of the 6475 age group

Were there as many males as females in the study group?

Line 118: non-asthmatic subjects showed a higher incidence of asthma in females than in males.

How can they show any incidence at all, when they were non-asthmatics?

The language is quite easy to read, but contains logical errors and some grammatical errors.

Author Response

October 6th, 2023

REVIEWER 2

Comments and Suggestions for Authors

The review “Phytoestrogen-based hormonal replacement therapy could benefit women suffering late onset asthma” aims to investigate if phytoestrogen is an effective treatment against late onset asthma in women. The aim of the study is recommendable and interesting. However, the aim is not addressed properly in the text.

Major concerns:

1: The manuscript is very unfocused. The aim of the review was to investigate whether phytoestrogen-based therapy decreases asthmatic symptoms in late onset asthmatic women. The main focus of the text should then be to investigate the prevalence of this patient group, what characterizes it and separates it from other groups, and if phytoestrogens would be beneficial by presenting clinical trials with these end points.

Investigating where phytosterols are found (foods, beverages), whether the active compounds are formed by the intestinal flora, what receptors are activated, confuse the point. Please revise the text to be more focused on the aim and less on other things, which of course are interesting, but do not further the argument.

Line 339: They may as well contribute to more efficient responses to infection and inflammation leading menopausal women to a much better quality of life.

This sentence illustrates the issue above. The aim was to investigate if phytoestrogen would be beneficial against asthma, not if it effective against infections or general inflammation or give better quality of life to menopausal women.

If you want to argue that phytoestrogens are a good treatment for asthma, you must show clear clinical evidence that this is the case. Explaining estrogen effects on transcription and GPCRs will not further the point.

The manuscript must be focused on the aim, or the aim must be changed.

Reviewer correctly states the following: “and if phytoestrogens would be beneficial by presenting clinical trials with these end points.” “…you must show clear clinical evidence that this is the case.”  We would be pleased to comply with the reviewer´s suggestions. Unfortunately, and precisely because of the lack of scientific papers in PubMed reporting clinical trials results regarding phytoestrogens beneficial effects on late asthmatic menopausal women, authors decided to write this review. Hopefully it will contribute to alert clinical researchers on the imperative need of developing proper studies in this regard.

On the other hand, authors truly consider that a theoretical proposal of complex biological issues and their probable interactions as is the case, should be described as extensively and as profoundly as possible. Even though the text seems to become unfocused, we believe that each section would promptly contribute to inform the interested reader on some of the main topics related to these intricate and subtly regulated pathophysiological issues. We appreciate and respect the reviewer´s point of view, but thoroughly disagree. In our opinion, the basic objective of a review is to inform and provide the readers with clues that could interest them and therefore would greatly regret shortening or modifying the actual text. We neither believe the title to be misleading nor wrong, since it is a proposal: “…could benefit women suffering late onset asthma”. Because of the former reasons and Reviewer´s 1 suggestion to extend the text further, we will not make substantial modifications to the actual text.

Nonetheless, we consulted clinicaltrials.gov on September 29th, 2023, and decided to add the following text to the manuscript to clarify this issue (Page 2, lines 56-71):

“Besides, published reports regarding phytoestrogens beneficial effects on late asthmatic menopausal women are practically non-existent. By consulting clinicaltrials.gov online, we found there is currently one study evaluating resveratrol/quercetin in the management of asthma, COPD and long lasting COVID (ID NCT05601180) [17], but no results have been published yet. Four clinical trials studying the implementation of isoflavones in asthma are reported: two were completed, one is recruiting study subjects and one is not yet recruiting (ID: NCT00277446, NCT01052116, NCT00741208, NCT05667701, respectively) [18-21].  None of these studies encompasses the special interest we discuss in our review. Interestingly, they evaluate inflammatory markers (exhaled nitric oxide, eosinophil LTC4 synthesis, eosinophil count, IL-6, serum CRP, and urinary leukotriene E4/creatinine ), pulmonary function (FEV1, FVC, peak flow, morning peak flow rate (PEFR)), symptomology, and asthma quality of life and control questionnaires (Asthma Control Test score, Asthma Symptoms Utility Index score, Marks Asthma Quality of LifeQuestionnaire score, Children’s Health Survey for Asthma score and Juniper Asthma Control Questionnaire). Conceivably, clinical trials evaluating phytoestrogens effects on late onset asthmatic women are urgently needed.”

2: Incorrect statements, for example:

Line 67: circulating levels of these inflammatory cells characterize atopic asthma and/or asthma severity

Is that strictly true? Are levels of circulating eosinophils directly correlated with asthma severity?

Some recent sources propose that circulating eosinophils counts are indeed related to asthma severity. Quote: “In eosinophilic asthma, eosinophils increase in the peripheral circulation and accumulate in the airway wall and the airway lumen; eosinophil activation and degranulation contribute to airway inflammation, mucus hypersecretion, mucus plugging, bronchoconstriction, and airway remodeling. Elevated blood eosinophil counts (BECs) are associated with more severe disease, increased frequency of exacerbations, and asthma mortality. Reducing eosinophil-associated airway inflammation is a therapeutic target for several asthma biologic agents. For patients with severe, eosinophilic asthma, biologic therapies that reduce or deplete eosinophils provide an endotype-specific treatment approach that results in significant reductions in asthma symptoms, reduction in oral corticosteroid (OCS) use, decreased exacerbation frequency, and improved lung function.” Taken from: Buhl R, Bel E, Bourdin A, Dávila I, Douglass JA, FitzGerald JM, Jackson DJ, Lugogo NL, Matucci A, Pavord ID, Wechsler ME, Kraft M. Effective Management of Severe Asthma with Biologic Medications in Adult Patients: A Literature Review and International Expert Opinion. J Allergy Clin Immunol Pract. 2022 Feb;10(2):422-432. doi: 10.1016/j.jaip.2021.10.059. Epub 2021 Nov 8. PMID: 34763123.

Nevertheless, pathological conditions are seldom strictly true.  Please take into account that this review only aspires to be a guide on the most recent and, to the authors ‘opinion, the most relevant issues that might contribute to the enhancement of the readers ‘criterion.

Line 120: this trend was observed throughout all the reproductive years

Please explain how “reproductive years” can be compared between males and females? Women reach menopause, men do not.

This phrase was misleading and now reads as follows: “…observed from puberty to menopause in women, and no…” Thank you for the suggestion.

Minor concerns:

1: There are many language errors in the text, for example:

First sentence of the abstract and line 37: what is “plasmatic concentrations”? Do you mean plasma concentration?

Yes, we mean plasma concentration. Plasma is a noun, while plasmatic is an adjective that qualifies the noun or subject, in this case “concentration”.

-Female sex steroid hormones periodical fluctuation

Not English, must be rephrased.

Thank you for your kind observation. The phrase was changed and now reads as follows: “Fluctuations in female sex steroid concentrations during menstrual periods are closely related to asthma symptoms…” Abstract, lines 18-19.

-relief for both their menopause

What is meant by that expression? How can you receive relief for a menopause? You can receive relief from symptoms due to menopause, but not from the menopause itself. Indeed, if you read the whole sentence, it states: “…relief for both their menopause and asthma symptoms”.  Abstract, line 25.

Line 35: symptoms in asthmatic women relate to their sex hormonal status

The following was added to the phrase to clarify the meaning “(i.e., perimenstrual period, pregnancy, menopause)”. Introduction, lines 34-35.

Line 45: asthma episodes frequency: please remove the final s in episodes. Done. Thank you for the observation.

2: Expressions with questionable meaning, for example:

warrants more thorough and comprehensive analysis.

The expression is too vague. Please specify what should be analyzed.

This expression now reads: “warrants future clinical studies that corroborate their beneficial value.” Introduction.  Page 2, lines 55-56.

This statement is wrong. Women do not suffer from asthma just because they are women. Please rephrase. I would if I could. Which statement? Page? Line?

Line 61: a considerable proportion of the female world population is prone to allergies and Th2-driven immunological responses

This is not limited to the female population.

Indeed. But literature cited in the manuscript (22-24 in the amended text) points out the fact that women are more prone to allergies and Th2-driven immunological responses than men. Phrase and citations are correct and in accordance to the context.

References in the Manuscript:

  1. Kanda, N.; Hoashi, T.; Saeki, H. The roles of sex hormones in the course of atopic dermatitis. Int J Mol Sci 2019, 20:4660. doi: 10.3390/ijms20194660
  2. Ridolo, E.; Incorvaia, C.; Martignago, I.; Caminati, M.; Canonica, G.W.; Senna, G. Sex in respiratory and skin allergies. Clin Rev Allergy Immunol 2019, 56:322-332. doi: 10.1007/s12016-017-8661-0
  3. González, D.A.; Díaz, B.B.; Rodríguez Pérez, Mdel. C.; Hernández, A.G.; Chico, B.N.; de León, A.C. Sex hormones and autoimmunity. Immunol Lett 2010, 133:6-13. doi: 10.1016/j.imlet.2010.07.001

Line 63: explain why asthmatic adult women have higher asthma prevalence than men (9.6% versus 6.3%, respectively) I cannot explain it. It is just a fact.

What do you mean by prevalence? Do they have more exacerbations?

Prevalence, as defined by the online edition of the Merriam-Webster dictionary, is “the percentage of a population that is affected with a particular disease at a given time”. This definition was added to the manuscript (page 2, line 79) to avoid any misunderstanding.

No, they do not suffer more exacerbations. They simply have the illness at a certain time. In the case of asthma more women (9.6%) of a certain population suffer the illness in comparison to 6.3% of men from the same population at the same time.

Line 112: It has been proposed that LOA develops between 27 ± 1.3 years (mean age at onset) and ≥65 years of age

What do you mean by this statement? Please clarify.

According to the sources cited in the manuscript (40, 44 in the amended text), late onset asthma begins when patients (mostly women) start showing symptoms at around 27 years of age in average, but this illness might develop any time between 27 and ≥65 years of age. Therefore, the ailment is categorized as “late onset”. To avoid any confusion, the phrase was reworded and now reads: “According to literature [40, 44] late onset asthma begins when patients (mostly women) start showing symptoms at around 27 years of age in average, but LOA might develop any time between 27 and ≥65 years of age,…” Page 2, lines 128-131.  

References in the manuscript:

  1. Miranda, C.; Busacker, A.; Balzar, S.; Trudeau, J.; Wenzel, S.E. Distinguishing severe asthma phenotypes: role of age at onset and eosinophilic inflammation. J Allergy Clin Immunol 2004, 113:101-108. doi: 10.1016/j.jaci.2003.10.041
  2. Gibson, P.G.; McDonald, V.M.; Marks, G.B. Asthma in older adults. Lancet 2010, 376: 803–813. doi: 10.1016/S0140-6736(10)61087-2

Line 114: The prevalence of asthma in adults older than 65 years is as high as 10%, and females are the main constituents of the 64–75 age group

Were there as many males as females in the study group?  

These data were obtained from actual Australian populations (Australian Centre for Asthma Monitoring. Asthma in Australia 2008. Canberra: Australian Institute of Health and Welfare; 2008), not from a controlled and properly designed study group. Conceivably, 10% asthma prevalence in adults older than 65 years of age points out that this ailment remains a health concern in the elderly. Even though we do not know the number of males in the whole population sampled, it remains relevant to recognize that most constituents of the 64-75 age group suffering asthma were females, a fact that casually coincides with similar data showing that asthma affects adult women more than men. In fact, epidemiology provides raw data, but is a valuable conceptual tool that should be used to design precise and useful clinical trials that would ideally lead us to unquestionable information.  

Line 118: non-asthmatic subjects showed a higher incidence of asthma in females than in males.

How can they show any incidence at all, when they were non-asthmatics?

Incidence, as defined by the online edition of the Merriam-Webster dictionary, is “the rate of occurrence of new cases of a particular disease in a population being studied”. This definition was added to the manuscript when used for the first time (page 3, line 126) to avoid any confusion.

So, they show incidence because they are new and show symptoms, therefore they are no longer non-asthmatics. They show symptoms may be because they suffer LOA.

Comments on the Quality of English Language

The language is quite easy to read, but contains logical errors and some grammatical errors.

Conceptual misunderstandings due to language or grammatical errors were corrected as well as possible. The amended version of the manuscript was proofread and reviewed by a native speaker. We hope that our upgraded manuscript now reaches the criteria of your high standard journal.

Round 2

Reviewer 2 Report

The review manuscript “Phytoestrogen-based hormonal replacement therapy could benefit women suffering late onset asthma” is a careful summary and comment on present literature concerning phytoestrogens to treat asthma as well as other lung conditions such as IPF. The text is well written and the references valid. There are only some minor concerns which need to be addressed:

Line 130: The text says “between 27 and ≥65 years of age”. The statement defies logic as there is no end year. From 27 onwards makes sense. Between 27 and 56 also makes sense. What point do the authors want to make?

Line 361: red clover Trifolium pratense L. Scientific names of plants are written in italics, Trifolium pretense.

Line 366: The text says “inflammatory markers expression”. The correct expression is “expression of inflammatory markers.

The language is generally good with no big issues. Please see specific comments for required language editing.

Author Response

October 11th, 2023

Reviewer 2

The review manuscript “Phytoestrogen-based hormonal replacement therapy could benefit women suffering late onset asthma” is a careful summary and comment on present literature concerning phytoestrogens to treat asthma as well as other lung conditions such as IPF. The text is well written and the references valid.

A: We appreciate your comment, thank you very much.

 There are only some minor concerns which need to be addressed:

Line 130: The text says “between 27 and ≥65 years of age”. The statement defies logic as there is no end year. From 27 onwards makes sense. Between 27 and 56 also makes sense. What point do the authors want to make?

A: We aggree with the observation and changed the sentence to “…from 27 years of age onwards”. Page 3 lines 130-131, typed in red.

Line 361: red clover Trifolium pratense L. Scientific names of plants are written in italics, Trifolium pretense.

A: Thank you. Correction was made. Page 8, line 361. Typed in red.

Line 366: The text says “inflammatory markers expression”. The correct expression is “expression of inflammatory markers.

A: Change was made. Thank you for the correction. Page 8, line 366. Typed in red.

Authors appreciate the Reviewer´s observations and hope that these modifications to our manuscript enhance it to meet your publication standars.